# Clinical Course and Ophthalmologic Findings in Idiopathic Intracranial Hypertension and Pregnancy

**DOI:** 10.3390/brainsci13121616

**Published:** 2023-11-22

**Authors:** Theresia Knoche, Leon Alexander Danyel, Lisa Varlet, Paula Haffner, Mohammad Salim Alzureiqi, Alexander Kowski, Verena Gaus

**Affiliations:** 1Department of Neurology, Charité Universitätsmedizin Berlin, 10117 Berlin, Germany; leon.danyel@charite.de (L.A.D.); lisa-justine.varlet@charite.de (L.V.); paula.haffner@charite.de (P.H.); alexander.kowski@charite.de (A.K.); verena.gaus@charite.de (V.G.); 2Department of Ophthalmology, Charité Universitätsmedizin Berlin, 10117 Berlin, Germany; m.alzureiqi@yahoo.com

**Keywords:** idiopathic intracranial hypertension, pregnancy, pseudotumor cerebri

## Abstract

Idiopathic intracranial hypertension (IIH) has its highest prevalence among women of childbearing age and therefore frequently coincides with pregnancy. This retrospective cohort study aimed to explore the impact of pregnancy on the clinical course, ophthalmologic findings and on the therapeutic management of IIH patients. Individual patient records were reviewed for neuro-ophthalmologic findings, treatment strategy, adherence to therapy and pregnancy complications. Sixteen patients with 19 documented pregnancies were identified. The visual acuity, visual field defects and the grade of papilledema at baseline and after pregnancy were compared. The visual acuity and visual field mean deviation at baseline and at follow-up after pregnancy did not significantly differ. Papilledema at baseline was more pronounced in patients who had been diagnosed with IIH during pregnancy than in patients with established IIH. In this cohort, the visual acuity and the visual field were not lastingly impacted by pregnancy. The adherence to therapy was low, with 69% discontinuing treatment or medication.

## 1. Introduction

Idiopathic intracranial hypertension (IIH) is a rare disease that is characterized by visual disturbance and disabling headaches due to raised intracranial pressure (ICP) of unknown etiology. It primarily affects obese women of reproductive age [1]. Apart from obesity, the dynamic of weight gain is a relevant risk factor for the disease [2]. As the diagnosis is usually made around 30 years of age, IIH inevitably coincides with pregnancy [3,4]. Available studies suggest that pregnancy may lead to manifestation of IIH [5,6] in any trimester of pregnancy, with 61% of patients being diagnosed in the first trimester [7,8]. In most of these women, IIH seems to resolve after the delivery, but it may recur during subsequent pregnancies. In patients who were diagnosed with IIH prior to conception changes in maternal physiology during pregnancy, such as weight gain, are proposed to aggravate the disease [5,6]. Pregnancy leads to an increase in abdominal pressure, which delays the venous return from the brain and may therefore result in increased intracranial venous pressure [8]. Furthermore, anemia is recognized as a risk factor for IIH in non-pregnant patients [9,10]. The physiological maternal anemia could thus also promote IIH. Hyper-estrogenemia during pregnancy is believed to worsen IIH, although this is not clearly established [11]. Lastly, Valsalva-induced increase in ICP during labor may negatively affect IIH symptoms [12,13].

Recently, the first prospective study investigating the outcome in pregnant women with IIH was published [14]. The results suggest that pregnancy does not adversely affect the visual outcome in IIH. Further, visual outcome in pregnant patients with established IIH and those who never had a pregnancy was comparable. Interestingly, the authors reported greater papilledema in patients who were first diagnosed with IIH during pregnancy than in pregnant patients with established IIH. The remaining literature on IIH in pregnancy is restricted to small retrospective studies and case reports, mostly published in the past century. Taken together, these investigations found no negative effects of pregnancy on the ophthalmologic outcome in IIH [7,8]. Conversely, IIH did not appear to adversely affect the course of pregnancy [15,16,17]. However, the available retrospective reports used heterogenous diagnostic criteria of the disease. Friedman et al. proposed novel diagnostic criteria for IIH in 2002, which were revised in 2013 [18,19]. These criteria have been accepted in current guidelines [20] and confirmed in clinical trials [21]. The defining features are a CSF opening pressure ≥25 cmH_2_O and the presence of papilledema next to a more rigorous exclusion of secondary causes to secure a definite diagnosis of IIH.

The treatment of IIH during pregnancy is complicated by the limited safety data of the available drugs. In addition, none of the recommended therapeutics have been approved by the U.S. Food and Drug Administration (FDA) for use in IIH (off-label therapy). With regard to pregnancy, it is important to note that topiramate is considered teratogenic due to higher rates of fetal abnormalities [22,23]. Acetazolamide has shown teratogenic potential in animals, but seems to be safe in humans, although evidence is limited [24]. Treatment with GLP-1 receptor agonists seems to be promising in the future, but is not routinely established yet, and cannot be recommended in pregnancy for lack of safety data [25]. Treatment guidelines recommend achieving disease remission or stabilization prior to pregnancy, as well as close patient monitoring throughout pregnancy to identify disease deterioration [26]. Patients are cautioned about excessive weight gain during pregnancy [20]. Because the data on IIH in pregnancy is restricted and the treatment options are limited during pregnancy, healthcare professionals are challenged to provide reasoned counselling and treatment to pregnant IIH patients. On the other side, women with IIH who desire to have children are conflicted by the remaining uncertainties. The aim of our study was to add to the existing evidence by investigating the clinical course and ophthalmologic findings of IIH in a retrospective cohort of pregnant patients with the diagnosis of IIH based on the revised Friedman criteria.

## 2. Materials and Methods

This is a single-center, retrospective cohort study evaluating the impact of pregnancy on the clinical course of IIH. Ethical approval was obtained from the local ethics committee. All methods were performed in accordance with local guidelines and regulations.

### 2.1. Patient Selection

A medical database inquiry identified patients treated under the definite or suspected diagnosis of IIH at our tertiary care center between January 2004 and October 2020. Individual patient charts were reviewed to determine whether the diagnosis of IIH could be verified based on the revised Friedman criteria for IIH [19]. According to the revised Friedman criteria, a diagnosis of IIH can be established in patients with papilledema and elevated lumbar puncture opening pressure (≥25 cmH_2_O) if alternate diagnoses have been excluded through adequate neurological examination, neuroimaging and cerebrospinal fluid analysis. Only patients with the definite diagnosis of IIH were included. Patients were excluded if data to determine the diagnosis of IIH was insufficient (i.e., lack of detailed fundoscopy or lack of MRI or measurement of CSF opening pressure). Patients with concurring neurological disorders affecting the central nervous system were excluded from this study. Patients were included for further analysis if documentation on pregnancies was available in their medical records. The individual patient data were reviewed in consideration of demographics (age, gestational age, and weight), medical history and clinical presentation, as well as clinical course of IIH before, during and after pregnancy. The ophthalmologic findings, IIH therapeutic strategy and the adherence to therapy were recorded. We assessed the outcomes and the possible complications over the course of pregnancy.

### 2.2. Pregnancy

Data on the course and outcome of pregnancy was extracted from neurological follow-up documentation, based on patient reported statements and, if available, obstetrical documentation.

### 2.3. Ophthalmologic Findings

Data was retrospectively extracted from reports of ophthalmologic examinations obtained during consultations at our hospital. Examinations were performed by an ophthalmologist and reviewed by a neuro-ophthalmologist. Best corrected visual acuity (BCVA) was transformed to logMAR for statistical analysis (logMAR = −log (decimal visual acuity). The BCVA in the more severely affected eye is reported in decimal values with logMAR values in parentheses. Visual field perimetric mean deviation (MD) in decibel [dB] was recorded on Humphrey visual field analysis using the 30-2° tendency-oriented perimetry. The extent of papilledema (mean papilledema grade = MPG) was extracted from ophthalmologic reports. Papilledema was graded according to the modified Frisén Scale [27] from grade 0 (normal optic disc), 1 (minimal degree of edema), 2 (low degree of edema), 3 (moderate edema), 4 (marked edema) to grade 5 (severe papilledema) from documentation of fundoscopic exams. If the eyes were affected differentially, the more severely affected eye was used.

In patients who were diagnosed with IIH during pregnancy, findings were each noted (1) at the time of IIH diagnosis and (2) after pregnancy. In patients with established IIH, ophthalmologic data was noted before and after pregnancy. If more than one ophthalmologic examination occurred before pregnancy, the examination closest to conception was used. If more than one ophthalmologic examination occurred after pregnancy, the earliest examination following delivery was used.

### 2.4. Statistical Analysis

All statistical analyses were performed with IBM SPSS Statistics software (IBM SPSS Statistics for Windows, Version 27.0. Armonk, NY, USA: IBM Corp.). The descriptive statistics are presented as mean ± standard deviation. Visual acuity in logMAR, the visual field defect MD and the grading of papilledema at baseline and after pregnancy were compared using paired t-tests for group comparisons and using Wilcoxon rank sum tests for non-parametric paired group comparisons. A two-tailed level of significance (*p*) was set to ≤0.05, 95% confidence intervals are reported for the difference of the means. GraphPad Prism (GraphPad Prism version 8.0.0 for Windows, GraphPad Software, San Diego, CA, USA) was used for graphic illustration.

## 3. Results

### 3.1. Patients

The medical database inquiry identified 243 consecutive patients who were treated between 2004 and 2020 and met the Friedman diagnostic criteria for IIH. Screening of the individual patient records identified 16 women (26.8 ± 5.7 years; range: 17–38 years) who had at least one documented pregnancy during IIH treatment; three women each had two pregnancies. Except for one woman, all were overweight, with a Body Mass Index (BMI) >25 kg/m^2^ at the time of IIH diagnosis. The BMI at the time of IIH diagnosis was 32.5 ± 8.0 kg/m^2^ (range: 20–48 kg/m^2^). Data on exact weight was missing in two patients. Three women (3/16 = 18.8%) had already had pregnancies prior to IIH diagnosis. Eleven women (11/16 = 68.9%) had been diagnosed with IIH before pregnancy. In the remaining five cases (5/16 = 31.3%), representing 2% of the IIH cohort (n = 5/243), IIH was first diagnosed during pregnancy, with IIH symptoms shortly preceding the diagnosis. Two of those women were diagnosed during the first trimester, and two were diagnosed during the second trimester. The gestational age at the time of IIH diagnosis was not documented in one woman. The demographic data and clinical findings are presented in Table 1.

### 3.2. Symptoms

All women reported visual disturbances (blurred vision, visual field defects) and all except one woman (94.8%) experienced headaches. Notably, two patients (12.5%) reported worsening of headache and visual symptoms during pregnancy which led to repeated presentations at the emergency department. Vertigo (n = 7, 43.8%) and tinnitus (n = 4, 25.0%) were less frequent. One patient presented with impairment of smell and taste. Six women (37.5%) reported intermittent diplopia. Among those, ophthalmologic examination identified one case of abducens nerve palsy and one case of trochlear nerve palsy. No objectifiable cause for diplopia was found in the remaining four women.

### 3.3. Therapeutic Strategy—Adherence to Therapy

None of the patients received or were recommended surgical therapy (i.e., shunting procedure, bariatric surgery) before or during pregnancy. Nine women were treated with acetazolamide before conception. Six of them (66.7%) were advised to stop the medication during pregnancy. Three women were prescribed acetazolamide through the course of their pregnancy, one of them declining treatment with acetazolamide (patient 15, Table 1). Patient 5 was asymptomatic under treatment with 1000 mg acetazolamide per day after suffering headaches and blurred vision before. Patient 16 presented with persistent headache, tinnitus and blurred vision, but had normal CSF opening pressure under medication with 375 mg acetazolamide per day. Five women were recommended repeated lumbar puncture with CSF extraction, which occurred regularly in two women and infrequently in the other two. One woman did not appear to the scheduled appointments for lumbar puncture (patient 3). Two women, who were both diagnosed with IIH during pregnancy, discontinued their follow-up appointments during pregnancy. Both later reported being asymptomatic after receiving the initial diagnosis of IIH until they presented again with recurrence of IIH symptoms independent from pregnancy several years (3 and 14 years) later. One woman repeatedly presented to the emergency room with headache and nausea, but declined further in-hospital treatment. In total, 10 women (10/16 = 62.5%) did not adhere to the recommended therapy, by either discontinuing the follow-up appointments (neurologic and/or ophthalmologic check-ups) or specifically refusing the advised treatment (CSF extraction or pharmacological therapy).

### 3.4. Ophthalmologic Findings

As reported above, most women were diagnosed with IIH prior to conception. In five patients, IIH was diagnosed during pregnancy. To improve comprehensibility throughout the text, we report baseline data for both groups. For the group of patients with established IIH before pregnancy, the term *baseline* refers to the dataset acquired before conception. For the other group, the term *baseline* refers to the ophthalmologic dataset acquired when IIH was first diagnosed. To evaluate changes in ophthalmological findings after pregnancy, baseline and follow-up datasets were compared. The term *follow-up* refers to the ophthalmological data available closest after pregnancy completion.

Each patient had received ophthalmologic assessment at the time of IIH diagnosis. All women had at least mild papilledema when IIH was first diagnosed. Follow-up data was available in 13 out of 16 patients (81.3%). The follow-up times ranged from 1 day to 13 years (mean: 3.8 ± 4.8 years, median: 2 years).

The MPG (mean papilledema grade) at baseline was documented in 13 women (13/16 = 81.3%). MPG at baseline was 1.7 ± 1.9 (median = 1) in all patients. In patients with established IIH, MPG at baseline was 0.8 ± 0.9 (median = 0.5, n = 9). The baseline MPG in women diagnosed with IIH during pregnancy was higher with 4.7 ± 0.6 (n = 4). After pregnancy, the overall MPG improved to 1.1 ± 0.9 (median = 1) compared to the MPG at baseline (mean difference = 0.6, 95% CI [0.6, 1.8]).

The BCVA at baseline was 0.92 ± 0.15 (logMAR: 0.05 ± 0.08, n = 15) in all patients. BCVA remained stable (meaning unchanged) in 6 out of 13 patients (46.2%). The follow-up BCVA was 0.95 ± 0.10 in all patients (n = 13, logMAR: 0.03 ± 0.05). The BCVA at follow-up did not differ compared to the BCVA at baseline (n = 12, *p* = 0.14, illustrated in Figure 1). In patients with established IIH, baseline BCVA was 0.97 ± 0.07 (n = 11, logMAR: 0.01 ± 0.03). The follow-up BCVA in patients with established IIH was 0.94 ± 0.12 (n = 8, logMAR: 0.03 ± 0.06), and did not significantly differ from baseline BCVA (difference of the mean: 0.03, 95% CI [0.06, 0.12]). In the subgroup of patients diagnosed with IIH during pregnancy (n = 4), baseline BCVA was significantly lower than in patients with established IIH (BCVA: 0.58 ± 0.21; logMAR: 0.27 ± 0.16, *p* = 0.0001, 95% CI [0.24, 0.54]). However, in patients diagnosed with IIH during pregnancy, the follow-up BCVA improved to 0.94 ± 0.09 (n = 5, logMAR: 0.03 ± 0.04) compared to baseline BCVA (difference of the mean 0.36, 95% CI [0.12, 0.60], *p* = 0.01). Figure 1 illustrates the BCVA in IIH patients at baseline and after pregnancy.

Data on visual field defects at baseline was available in 11 patients (11/16 = 68.8%). Eight women had no relevant visual field defects. Three women had scotoma with visual field MD of 14.5, 15.6 and 7.2 dB in the more severely affected eye, respectively. The follow-up data on visual field MD was available in 12 patients. Four women had no visual field defects on follow-up. Three had mild scotoma with MD of −3.0, −2.8 and −3.4 in the more severely affected eye, respectively. The remaining five patients had scotoma with MD of −5.0, −10.8, −5.2, −5.1 and −15.9. Taken together, comparing visual field MD for the more severely affected eye before and after delivery showed no significant worsening (n = 7, *p* = 0.5).

### 3.5. Pregnancy Outcomes

Data on the pregnancy outcomes was not available in five pregnancies (5/19 = 26%). No fetal complications during pregnancy and delivery were reported in 12 out of 19 (63%) pregnancies. For one woman (patient 10), two abortions were reported: one medically induced due to worsening of IIH-symptoms in the eighth gestational week and one spontaneous abortion in the seventh gestational week of her second pregnancy. Data on the method of delivery was available in four pregnancies with three primary Caesarean sections and one spontaneous delivery.

## 4. Discussion

IIH is characterized by intracranial hypertension of undetermined source. If untreated, patients may suffer from persistent headache and lasting visual impairment due to development of diplopia and/or scotoma [28,29]. The disease is most prevalent among women of childbearing age and may therefore frequently coincide with pregnancy. Concerns regarding the potential worsening of IIH symptoms and findings during pregnancy arise, especially because of the limited safety data on the available pharmaceuticals and due to uncertainties concerning the treatment strategy.

This study evaluated the impact of pregnancy in a retrospective cohort of 16 IIH patients applying the revised Friedman criteria for the diagnosis of IIH. Most patients were overweight or obese (94%), with obesity being a key risk factor for the development of IIH [2]. Two women who were first diagnosed in pregnancy had a BMI < 30 kg/m^2^ at the time of diagnosis in the first and second trimester, respectively, indicating that they were not obese before pregnancy. In our cohort, the prevalence of pregnancy during IIH was 6%, which is within the 5% to 15% previously described [7,8]. Consistent with the literature, our patients primarily developed symptomatic IIH within the first two trimesters [7,8]. It was previously assumed that pregnancy may provoke or exacerbate IIH [30,31,32]. Contrarily, Giuseffi et al. [33], Digre et al. [6] and a larger Swedish register study [34] with 902 IIH patients found no correlation between pregnancy and the diagnosis of IIH. In our investigation, pregnant IIH patients represented 6.6% of the total IIH cohort (16/243). Due to the retrospective study design, we cannot exclude information bias in terms of underreporting of pregnancies because of missed or induced abortions. However, only five (2%) of the 243 IIH patients were first diagnosed with IIH during pregnancy. Our data does not suggest a causative association of IIH and pregnancy. We speculate that IIH more likely affects pregnant women by chance, as IIH is most prevalent in women of childbearing age. Consequently, IIH symptoms in our cohort were comparable to those of non-pregnant women.

### 4.1. Ophthalmologic Findings

In this cohort of IIH patients we did not observe a persistent worsening of visual acuity and visual field defects after pregnancy. However, in the subgroup of patients diagnosed with IIH during pregnancy, the visual acuity at baseline was lower than in patients with established IIH. Likewise, patients with newly diagnosed IIH had more severe papilledema at baseline than patients with established IIH. Both visual acuity and papilledema grading improved at the time of follow-up. A recent prospective study reported the clinical outcomes of 51 IIH patients who had been pregnant during IIH treatment (46 patients with established IIH, six patients newly diagnosed with IIH during pregnancy) and a control group of 325 IIH patients who had not been pregnant or who had pregnancies prior to IIH diagnosis [14]. The visual outcome was comparable to IIH patients who had never been pregnant. The authors reported greater papilledema in patients who were first diagnosed with IIH during pregnancy than in pregnant patients with established IIH, a finding we also observed in our cohort. The remaining literature on IIH and pregnancy is restricted to single cases or small case series, both reporting patients who were either diagnosed with IIH during pregnancy or who became pregnant after IIH diagnosis. Taken together, the existing retrospective studies also suggest that pregnancy is not likely to worsen the clinical course of IIH [6,11,16]. Huna-Baron and Kupersmith reported 16 pregnancies in 12 women, in whom visual outcome was the same as in non-pregnant patients [7]. Digre et al. reported no demographic and clinical differences between pregnant and non-pregnant women by comparing pregnant IIH patients to an age-matched control group [6]. These retrospective studies used varying inclusion criteria, mostly because of repeated redefinitions of IIH since its first description. Advances in neuroimaging and the identification of causes of secondary intracranial hypertension have led to a more precise definition of the condition now referred to as IIH, highlighting the need for studies with well-defined IIH cohorts.

Despite pregnancy leading to weight gain and despite delivery and bearing down leading to a transient increase in ICP, the here-reported women had no lasting visual impairment after pregnancy. We cannot state if transient visual impairment occurred during pregnancy in women diagnosed with IIH prior to pregnancy, because the ophthalmologic data during pregnancy itself was too cursory.

### 4.2. Treatment and Adherence to Therapy

In our cohort the overall adherence to the pharmacological treatment and non-pharmacological therapy (regular follow-up visits and CSF extraction) over the course of pregnancy was surprisingly low: A total of 10 women (10/16 = 62.5%) did not adhere to the recommended therapy by either discontinuing the follow-up appointments or specifically refusing treatment. Although it is regarded as a safe procedure during pregnancy, the prospect of repeated lumbar punctures, which are known to be traumatizing [35,36], may in part account for the low adherence. The overall utility of repeated CSF extraction is being debated [37]. Only two women in our cohort received pharmacological therapy for IIH. They were prescribed acetazolamide, with no reports on adverse effects on the course of pregnancy, albeit no systematic workup regarding fetal or maternal adverse effects was available. One woman was recommended treatment with acetazolamide but declined. Although prescription of acetazolamide during pregnancy is possible, depending on an individual evaluation of risks and benefits [26], the overall safety-data is limited and most of our patients were recommended non-pharmacological treatment.

### 4.3. Limitations

Our study is limited by its retrospective design. Therefore, standardized follow-up times were not available and follow-up ophthalmologic data was missing in some patients. Additional optical coherence tomography (OCT) remains to be desired, but OCT was conducted in only a few patients and was therefore not included in the analysis. Furthermore, our data on pregnancy outcomes and on patient reported outcomes was only limited and the reasons for the low adherence rate remain unknown. However, the findings from our retrospective cohort add new aspects to the existing data on the course of IIH during pregnancy and provide a basis for future investigations.

## 5. Conclusions

Taken together, the findings of our retrospective investigation on pregnant women with IIH indicate that (1) IIH does not seem to be triggered by pregnancy, (2) there appears to be no lasting visual impairment after pregnancy, although (3) therapeutic adherence is compromised. As papilledema may be more severe in patients with newly diagnosed IIH in pregnancy than in those with established IIH, this group of patients should be monitored especially closely. Treatment guidelines recommend a multidisciplinary therapeutic approach including neurologists, ophthalmologists, and obstetricians to optimize the care for IIH in pregnancy [19].

## Figures and Tables

**Figure 1 brainsci-13-01616-f001:**
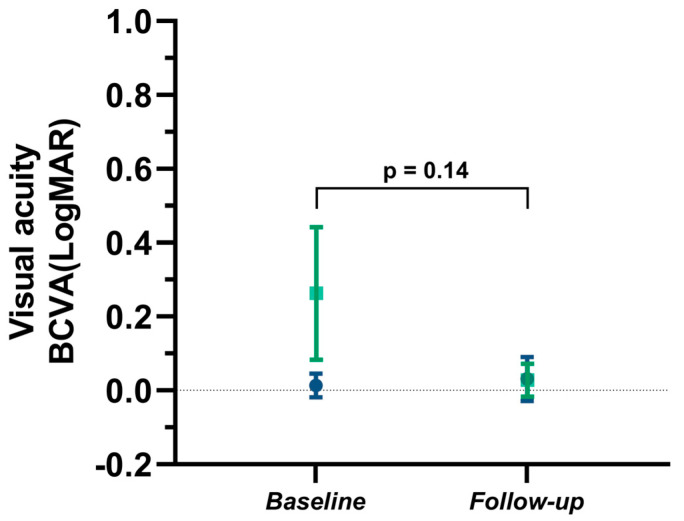
Comparison of BCVA in IIH patients at baseline and after pregnancy. The mean BCVA (logMAR) at baseline and at follow-up is shown for patients with established IIH (blue boxplots) and for patients newly diagnosed with IIH during pregnancy (green boxplots). BCVA = best corrected visual acuity.

**Table 1 brainsci-13-01616-t001:** Detailed patients’ characteristics including demographics, the treatment regimen, the adherence to therapy and ophthalmologic findings for each patient during each pregnancy.

Patient	PY	Age	BMI	BCVA Baseline	Visual Field MD Baseline	Treatment	ATT	Pregnancy Outcome	BCVA Follow-up	Visual Field MD Follow-up
1	1	37	34	1/1	0/0	LP, clinical follow-up	+	completed	1/1	3/2.5
	2	42		1/1	3/2.5		−	N/A	N/A	N/A
2	1	38	38	1/1	7.2/7	clinical follow-up	+	completed	1/1	8.7/10.7
3	1	28	26	0.6/0.7	N/A	LP	−	completed	0.9/0.9	5/4.4
4	1	24	48	N/A	N/A	clinical follow-up	−	completed	1/1	2.7/5.2
5	1	30	38	1/1	N/A	acetazolamide	+	completed	1/1	2.8/2.7
6	1	18	28	1/1	0/0	clinical follow-up	−	completed	1/1	0/0
7	1	23	34	1/0.8	0/0	clinical follow-up	−	completed	1/0.8	1.3/5.1
	2			1/0.8	1.3/5.1	clinical follow-up	−	N/A	N/A	N/A
8	1	27	28	1/1	0/0	clinical follow-up	+	completed	1/1	0/0
9	1	26	28	1/0.8	N/A	LP	+	completed	1/1	0/0
10	1	23	32	0.3/1.25	central scotoma	LP, induced abortion	+	induced abortion	0.8/1.25	enlarged blind spot
	2	37		0.7/1.0	14.5/3.4	clinical follow-up	+	missed abortion	N/A	N/A
11	1	24	27	1/0.9	0/0	clinical follow-up	−	completed	1/1	1.6/2.0
12	1	17	N/A	0.6/1	N/A	clinical follow-up	−	N/A	1/1	3.4/2.2
13	1	27	33	1/1	0.5/0.4	clinical follow-up	−	N/A	N/A	N/A
14	1	32	46	1/1	0/0	clinical follow-up	−	N/A	N/A	N/A
15	1	26	20	1/1	0.4/1.4	acetazolamide	−	completed	1/1	N/A
16	1	29	26	0.8/0.9	15.6/10.5	acetazolamide, LP	+	completed	0.7/0.9	15.9/11.9

Patients No. 3, 4, 9, 10 and 12 were newly diagnosed with IIH during pregnancy, their respective rows are marked with a gray background. The remaining patients had established IIH before pregnancy. Abbreviations: ATT = adherence to therapy, BMI = body-mass-index [kg/m^2^], BCVA = best corrected visual acuity, decimal values, (right/left), PY = pregnancy, LP = lumbar puncture with therapeutic CSF extraction, MD = mean deviation in [dB], right/left, N/A = data not available.

## Data Availability

The datasets generated for the current study are not publicly available but are available from the corresponding author on reasonable request.

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
