# Peer review of "Clinical Course and Ophthalmologic Findings in Idiopathic Intracranial Hypertension and Pregnancy"

_brainsci, 2023, doi:10.3390/brainsci13121616_

Round 1

Reviewer 1 Report

Comments and Suggestions for Authors

This is a nice paper looking at women with a IIH diagnosis exposed to pregnancy. This is a common clinical question that appears for clinicians treating patients with IIH. Although it is not a high prevalence of patients with IIH and are exposed to pregnancy, IIH is becoming more prevalent . As IIH primarily affects females in the reproductive years it is important that we have some scientific based knowledge to rely on to give correct advice to females with IIH and their health providers. 

This paper is well structured and easy to follow regarding used method and the results. It is important to present structured data on what happens to women with IIH during pregnancy although the sample size due to the rareness of disease tends to be small. 19 pregnancies therefore is a fair sample size. 

A minor comment - Table 1: I read in the Table 1-text that patients newly diagnosed with IIH during pregnancy is supposed to be marked with a gray background, however this is not seen in the version that I received - I guess that will be done and checked before submission - as it would greatly help the reader. 

Your findings that pregnancy does not seem to predispose/trigger IIH development are consistent with the literature. I´d like to address that your findings are also consistent with those of a large Swedish diagnosis register study on IIH patients (902 patients) compared to matched controls (4510 matched general populations controls + 4510 matched general population controls with an obesity diagnosis) where 7% in all three groups had exposure to at least 3 month exposure to pregnancy the year prior to index date (first date of diagnosis of IIH (same date used for matched controls). This study does not have information on severity and outcome but from an epidemiological view with a larger cohort it strengthens the evidence that pregnancy exposure does not seem increase the risk of IIH as the exposure to pregnancy is consistent with that of the general population. (Sundholm et al. A national Swedish case-control study investigating incidence and factors associated with idiopathic intracranial hypertension. Cephalalgia 2021). 

Author Response

Dear reviewer,

thank you kindly for taking the time to review our manuscript. We greatly appreciate your comments and your mentioning of the Swedish register study, as it supports our findings. We have included their results in the discussion paragraph. You will find the changes to the paragraph highlighted in the re-submitted manuscript file.

Regarding comment 1: Please excuse the mistake with the missing gray background in table 1, this was corrected in the re-submitted manuscript file.

Kind regards,

the authors.

Reviewer 2 Report

Comments and Suggestions for Authors

This work outlines the evolution of IIH and its ophthalmological impact
throughout pregnancy. I believe it is of value to ophthalmologists and other
specialties alike.

The sections are well organized; The work is well researched and comprehensive in its approach; However, the number of patients is quite low and the sample is heterogenous. It is difficult to draw conclusions, this study has rather descriptive quality

- Please clarify and detail in Patient Selection the inclusion and exclusion criteria for the study (including restating the Friedman criteria)- Please state the Papilledema Grading System (Frisen Scale)

- Linked t-test - I believe the correct and widely used term is paired t-test

- Line 149 - gray background not visible

- There are several dated references - I strongly suggest updating references no. 2-5, 7, 13, 25, 26, 28, 29-31, and suggest updating any references older than 10 years

Comments on the Quality of English Language

Please rephrase and correct words in green (severer, funduscopic, abducent)

Author Response

Dear reviewer,

thank you very much for taking the time to review our manuscript. We greatly appreciate your feedback and your comments. Please find the detailed responses below and the corresponding corrections highlighted in the re-submitted manuscript file.

Comment 1: Please clarify and detail in Patient Selection the inclusion and exclusion criteria for the study (including restating the Friedman criteria)- Please state the Papilledema Grading System (Frisen Scale).

Response: We updated the inclusion and exclusion criteria in the patient selection paragraph, Friedman criteria were specified. The changes are highlighted in yellow. A more detailed description of the modified Frisen Scale was added to the methods paragraph.

Comment 2: Linked t-test - I believe the correct and widely used term is paired t-test.

Response: Thank you for pointing this out to us, the term was adjusted to “paired t-test” in the manuscript file.

Comment 3: Line 149 - gray background not visible

Response: We are sorry for the mistake with the missing gray background in the table. We corrected the issue.

Comment 4: There are several dated references - I strongly suggest updating references no. 2-5, 7, 13, 25, 26, 28, 29-31, and suggest updating any references older than 10 years

Response: Due to inclusion of new references, the current reference list differs from the one in the former manuscript. Here, we refer to the numbers you mentioned in the former version manuscript:

    • We updated references No. 2, 3, 4, 25, 26.
    • Updating reference No. 5 and 7 would result in exclusion of reporting a proportion of the existing data (which is still sparse) to base recommendations for IIH and pregnancy.
    • Regarding reference No. 13: It is the first publication to report a connection between ICP and Valsalva, therefore we would prefer not to discard it. We added a more recent publication, which also proves the relationship.
    • References No. 28-31 refer to the previous notion, that pregnancy was associated with IIH. From our clinical experience, this belief (just like the references) is outdated. However, this assumption is still often picked up by colleagues or patients, which is why we do not want to leave it out. It has not been proven by newer studies, which are also cited. Therefore, we believe that in this case the older references are useful.

Comment 5: Please rephrase and correct words in green (severer, funduscopic, abducent)

Response: Unfortunately, we were not able to see the green markup you mentioned. We hope that our correction is still sufficient. Please let us know if we still missed it.

Kind regards,

the authors.

Reviewer 3 Report

Comments and Suggestions for Authors

This is an interesting study that, albeit retrospectively, highlights the influence of pregnancy on the course of IIH. The methodology of the study involved 243 consecutive subjects with IIH during pregnancy and how this changed the parameters of IIH. Sixteen women from this group of subjects with IIH had one or more pregnancies (n=19).

The results indicate that IIH does not appear to be triggered by pregnancy and there does not appear to be any lasting visual impairment after pregnancy, although adherence to treatment is suboptimal. Mail papilledema may be more severe in patients with newly diagnosed IIH in pregnancy than in those with established IIH, this group of patients should be monitored with particular attention by a multidisciplinary team, which unfortunately does not happen on a regular basis.

I suggest considering the following papers for a more granular definition of IIH.

https://doi.org/10.1186/s10194-023-01641-x

https://doi.org/10.1186/s10194-023-01631-z

Author Response

Dear reviewer,

thank you very much for taking the time to review our manuscript. We greatly appreciate your consideration. Thank you for mentioning both papers to us and we agree that they support a more defined definition of the disease. We included references to the introduction paragraph.

Kind regards,

the authors.